# Automatic Pavement Defect Detection and Classification Using RGB-Thermal Images Based on Hierarchical Residual Attention Network

**DOI:** 10.3390/s22155781

**Published:** 2022-08-02

**Authors:** Cheng Chen, Sindhu Chandra, Hyungjoon Seo

**Affiliations:** 1Department of Civil Engineering, Xi’an Jiaotong-Liverpool University, Suzhou 215123, China; cheng.chen19@student.xjtlu.edu.cn; 2Department of Civil Engineering and Industrial Design, University of Liverpool, Liverpool L69 3BX, UK; s.chandra@liverpool.ac.uk

**Keywords:** pavement defect classification, deep learning, hierarchical residual attention network, attention mechanism, visual interpretation

## Abstract

A convolutional neural network based on an improved residual structure is proposed to implement a lightweight classification model for the recognition of complex pavement conditions, which uses RGB-thermal as input and embeds an attention module to adjust the spatial, as well as channel, information of the images. The best prediction accuracy of the proposed model is 98.88%, while the RGB-thermal is used as input and an attention mechanism is used. The attention mechanism increases the attention to detail of the image and regulates the use of image channels, which enhances the final performance of the model. It is also compared with state-of-the-art (SOTA) deep learning models, indicating our model has fewer parameters, shorter training time, and higher recognition accuracy compared to existing image classification models. A visualization method incorporating gradient-weighted class activation mapping (Grad-CAM) is proposed to analyze the classification results, comparing the data the model learns from the images under different input data.

## 1. Introduction

With the growth of city traffic and the resulting increase in traffic volume over recent years, the timely maintenance of paved roads has become very important. Pavements can be damaged due to the effects of temperature change and climate change [1]. Water penetration into pavement cracks can exacerbate damage, resulting in structural defects in pavements [2]. Damage to the pavement not only affects the safety of the driver but can also cause casualties due to traffic accidents. Therefore, immediate maintenance is required for safe road operation.

Currently, pavement crack detection is performed manually, and repair mainly involves filling the cracks with sealant. Automated pavement detection systems have been studied, but previous studies have mainly focused on crack extraction. However, for complex road conditions, existing methods are limited in identifying elements in the pavement, including cracks [3]. Multi-sensor fusion processing ideas for complex pavement conditions have been studied with technologies such as acceleration sensors [4], infrared sensors [5], and multi-vision cameras [6]. 3D laser scanning can also provide additional identification information to optical images of roads [7]. Therefore, studies that identify damage to the pavement through various sensors have been conducted as a flow of research. This study also has limitations in that it is not possible to inspect all pavements at the national level. 3D laser scanning has also been used recently to detect the displacement and damage of infrastructure [8,9,10,11,12,13]. Zhao et al., (2022) used a machine learning algorithm to detect the movement of the retaining structure [14].

Additionally, there has been a shift from manual to automatic pavement defect detection due to the advancements in computer vision and artificial intelligence (AI). Automatic detection systems are expected to identify different pavement cracks quickly under various conditions, even under adverse weather conditions [15]. However, because a single sensor cannot handle the challenging conditions, and a basic detection model also cannot handle the redundant data from several sensors, it raises growing challenges for defect detection modeling [16]. To this end, data acquisition requires the selection of appropriate sensors to quantify the pavement damage. Acquisition systems based on optical devices are easier to develop and operate than any other type of system (vibration, LiDAR (light detection and ranging)) due to their reduced sensitivity to motion and vibration. However, the use of optical devices alone cannot eliminate the effect of factors such as illumination on the accuracy of quantifying pavement damage. Multi-sensor synergy can improve and compensate for the lack of information from a single sensor [17]. Kim et al., [18] introduced a new deep learning framework for camera and LiDAR sensor fusion. Chen et al., [19] proposed a simple fusion method using 50% transparency of thermal images and RGB images, which effectively improves recognition accuracy. Thermal imaging tends to have better real-time efficiency and lower costs [20]. Thermal images are also used for pavement crack detection, where the surface temperature distribution pattern is directly related to the pavement crack profile and can be used as an indicator of crack depth [21].

Another trend for crack detection in road pavements is to analyze raw data using algorithms such as deep learning and machine learning as data analysis technology advances. For automatic pavement detection, studies using acceleration data and studies using images are in progress [19,22]. Image processing achieves image recognition through machine learning, and machine learning can be divided into conventional machine learning and deep learning [23]. Image processing can be required to remove various elements of the pavement from the image data [24]. Zou et al., proposed a shadow removal algorithm before crack extraction to remove the effect of shadows on road surface detection [25]. However, crack recognition requires the support of some machine learning algorithms, such as Support Vector Machine (SVM), Radial Basis Function (RBF), K-Nearest Neighbor (KNN), and Random Decision Forest [26,27]. Principal component analysis (PCA) was used in conjunction with machine learning to speed up single-crack identification [22]. Xu et al., [28] aimed to assess pavement damage by extracting cracks and potholes, classifying them from vision. Due to their better generalization and portability, convolutional neural networks (CNNs) have been gradually applied to extract crack features in combination with image processing methods [29]. Sholevar et al., [30] proposed a deep convolutional neural network (CNN) as a detection system for asphalt pavement cracks, which is capable of robust detection and classification of pavement cracks. Li et al., [31] proposed a new method to automatically classify image blocks cropped from 3D pavement images using a CNN model. A deep learning-based supervised method with the ability to handle different pavement conditions is proposed [32]. An autonomous measurement scheme is introduced to collect, analyze, and map image-based distress data in real-time [33]. The stability and superiority of the optimal algorithm were verified through testing and comparison studies [34]. To address these issues, an instance segmentation network for pavement crack detection was proposed [35].

With the advancement of deep learning, the more useful sensor data and deeper and wider networks mean that more information can be obtained, but at the same time, it may make the input information redundant and invalid, which may reduce the performance of the model. Hu et al., [36] proposed SENet (Squeeze and Excite Network) to try to solve this problem. SENet uses a new feature recalibration strategy to automatically obtain the importance of each feature channel. Woo et al., [37] proposed CBAM (Convolutional Block Attention Module), which is a feed-forward convolutional neural network that combines spatial and channel attention mechanisms. Since it is a lightweight module, it can be integrated into a convolutional neural network structure to enhance the performance of the multi-sensor detection model.

Considering the limitations of existing pavement defect detection methods based on optical images, such as the susceptibility to uneven illumination and the false detection rate for complex road conditions, an effective CNN framework is proposed to process RGB-thermal images for pavement defect classification of complex pavement conditions. Deep learning networks based on hierarchical residual networks and attention mechanisms are proposed, inspired by hierarchical residual networks for effectively extracting multiscale features and attention mechanisms for reassigning weights to multi-sensor data. The proposed framework merges residuals into the network, which solves the problem of increasing computational complexity and the number of parameters due to the stacking of convolutional layers in traditional convolutional neural networks. The proposed model is trained to learn one weight per input to emphasize important features and ignore irrelevant features, which assigns a weight to the input information from multi-sensors, thus filtering out information that is not effective for classification performance. In addition, to visualize the most typical feature information, we compute an attention map by mapping with a re-weighted depth feature map. The results of the AvgPool and MaxPool layers in the attention module are fed to two 1D convolutions and a kernel-size value-adjusted 1D convolution to qualitatively analyze the effectiveness of the network classification. This paper is structured as follows: The research dataset in this study is described in Section 2. Section 3 provides a specific description of the proposed model. Section 4 describes the main results of our study. In Section 5, we discuss the limitations of the study and possible further work, followed by the conclusion section (Section 6).

## 2. Dataset and Preprocessing of the Thermal Image

All pavement distress data were obtained in Liverpool, UK. Nine pavement distress types were considered, i.e., transverse cracks, longitudinal cracks, alligator cracks, joints or patches, potholes, manholes, shadows, road markings, and oil markings. The FLIR ONE camera connected to a phone is used for data acquisition, where the FLIR ONE’s built-in application displays real-time thermal infrared images and acquires both RGB and thermal images. The images were subsequently extracted through the MATLAB API interface provided by FLIR. Figure 1 shows RGB, thermal, and its fusion image samples of the pavement defects and detectable markers. Each image is captured separately to the RGB (red, green, blue) image and the thermal image. The fusion image has four layers (Red, green, and blue) and is obtained by overlaying the RGB and thermal image and used in the subsequently proposed network. Usually, the RGB image shows the texture and color characteristics of the pavement defects and markers, while the thermal image shows the temperature differences, which also reflect the material and depth differences. Infrared acquisition data are easily affected by ambient temperature, and this experiment uses a passive heat source for acquisition, so the collected temperature data are changed by the seasonal influence. To eliminate the effect of ambient temperature, the thermal image preprocessing in Section 3.2 is described in detail. Since the RGB images and thermal images were acquired simultaneously by FLIR ONE, both images have been subjected to an in-instrument-based alignment. After completing the data collection, the images were manually classified and labeled.

As shown in Figure 1, the characteristics of the thermal and optical images can be found, and the red color in the thermal image has a higher temperature. It is obvious from Figure 1 that the crack area in thermal images is more obvious than in RGB images. On the contrary, the texture images of the cracks are not obvious in the RGB images. Areas with asphalt or oil stains are more visible in the thermal image, which applies to joints and oil marks, because asphalt and oil materials have better heat absorption properties with a heterogeneous temperature difference. On the other hand, the manhole areas have lower temperatures than the normal pavement areas because of the general separation between them. In addition, the potholed areas seem to have higher roughness and slower heat absorption properties, which leads to a lower temperature distribution. Pavement markings (usually white or yellow markings) show lower temperatures (than normal pavement) due to their low heat absorption properties. Finally, they show a lower temperature distribution than normal pavement due to the shadows blocking the sunlight. The data are tagged with the category at the time of collection, and the tagged data are collected in different folders. The number of each distress type in the dataset is shown in Table 1.

## 3. Methodology

### 3.1. Overview of the Proposed Model

A hierarchical residual network with an attention mechanism was built to complete the pavement defect classification. The overall architecture of our proposed network is shown in Figure 2. To fully highlight the spatial information of images and to also better extract information from the RGB images and thermal images, a hierarchical residual architecture was used in the proposed model. Specifically, the front-end structure with the hierarchical residual block took the image as input to extract features, which include the R, G, B, and thermal channels of the pavement image. Since the value of the thermal image is the extracted temperature information, it is not applicable to the convolution operation. The temperature information was mapped to the pixel grayscale value of [0, 255]. The convolution operation in the network and the hierarchical residual block were used to extract the features. The attention module recalibrated the spatial features at different scales, and also assigned different weights to the features of the images from different sensor inputs. The fused adaptive weighted channels and spatial features were input into the classification module, which can identify the pavement defect more finely and filter the feature information of non-defect parts to improve the accuracy of model recognition. Finally, the SoftMax layer was used to calculate the classification probability for each defect type. The different components of the proposed approach are described in detail in the following section. Table 2 shows the relevant parameters of the proposed model.

### 3.2. Thermal Transfer to Grayscale Image

In order to facilitate the convolution of the thermal image in the model, the temperature of the thermal image was mapped to the pixel grayscale value of [0, 255]. As shown in Equation (1), the temperature interval is $\max\left(T\right)-\min\left(T\right)$, after scaling the temperature mapping to [0, 1]. Then, it was multiplied by 255 to obtain the thermal grayscale image. The result is shown in Figure 3. The grayscale map reflects the same information details as the original thermal image, which is more convenient for the operation of convolution.
(1)$$grayscale=\frac{T-\min\left(T\right)}{\max\left(T\right)-\min\left(T\right)}\times 255$$

### 3.3. Hierarchical Residual Block

For traditional deep learning networks, we generally believe that the deeper the network (more parameters), the better the nonlinear representation that the network can learn. When increasing the number of network layers, the network can perform more complex feature pattern extraction, so better results can be achieved, in theory, when the model is deeper. However, as the depth of the network increases, the accuracy reaches saturation and then decreases rapidly [38]. This network degradation problem makes it difficult to train deep learning models because more layers lead to larger training errors, causing gradients to disappear. The key idea of residual learning is to introduce constant mapping into the backbone path of the network structure [39]. In other words, if the number of layers is deepened, instead of simply stacking more layers, one layer is stacked so that the output after stacking is the same as the output before stacking to ensure identity mapping. In the training process of the deep residual network, the underlying error can be propagated through a shortcut, which can effectively solve the gradient disappearance problem. Residual learning does not require additional parameters, so it neither adds additional parameters nor increases the computational complexity compared with the original network [40].

Based on the residual learning network, extracting multi-scale features is crucial for the image classification task, and can also effectively add multiple available receptive fields to expand the feature extraction under different fields of view. Thus, it will increase the diversity of features to improve the prediction accuracy of the network. Figure 4 illustrates the general structure of the residual and hierarchical residual blocks. The hierarchical residual block is updated from the residual block. The hierarchical residual block divides the input feature maps into several groups, and the feature maps of each subgroup are executed by different layers of the convolution operator. In the layered residual block, different subgroups of the feature maps have different perceptual fields, so the combined feature maps can represent multi-scale features and therefore increase the perceptual field of the network. Figure 4b shows a hierarchical residual cell with 4 scales, where ⊕ denotes the connection operation.

### 3.4. Attentional Mechanisms

In cognitive science, due to bottlenecks in information processing, humans selectively attend to a portion of all information while ignoring the rest of the visible information. Inspired by these, attention mechanisms can recalibrate input features by explicitly establishing input weight assignment relationships. Traditional classification models using a single sensor as the input assign equivalent weights to all pixels and RGB image channels separately. In fact, different spatial pixels contribute unequally to the discrimination of the classification results, while the inputs from different sensors contribute unequally to the discrimination of the classification results. For example, these interfering pixels can weaken the discriminative power of spatial-channel features and thus affect the classification accuracy. If the weights of these pixels or channels that are detrimental to classification can be suppressed, the discriminability of spatial-channel features will increase. Therefore, it is feasible and beneficial to introduce an attention mechanism into the classification of multi-sensor data as input, which can focus more on discriminative valid spatial and channel features and weaken the unfavorable information for classification. Therefore, we employ both the channel attention module and spatial attention module to recalibrate channel and spatial features at multiple scales. The attention module used in this paper is derived from CBAM (Convolutional Block Attention Module) [37], as shown in Figure 5. Attention can be performed in both channel and spatial dimensions. The two attention modules are described in detail below.

The features input into the attention module are generally C×H×W, where C is the number of channels and W and H are the width and height of the reduced image obtained after convolution, respectively. The application of channel attention is to assign different weights to the channels of the image to improve the representation capability of the model, and in this paper, channel attention weights indicate whether to focus on or ignore RGB-thermal channels. The structure of the channel attention module is shown in Figure 6. Channel attention emphasizes reducing channel redundancy and constructing channel attention graphs by capturing the inter-channel relationships of the features. Given a feature map in the input layer, feature squeezing and aggregation are handled by average and max pooling, which are executed simultaneously to produce two different feature maps. Then, they are fed into a shared network consisting of two dense layers for training. In summary, the channel attention is calculated as follows:(2)$$MC=\sigma \left(FC\left(Max_{pool}{}_{\left(X\right)}\right)+FC\left(Avg_{pool}{}_{\left(X\right)}\right)\right)$$
where σ denotes the sigmoid function and $Max_{pool}$ is the maximum pooling operation, while $Avg_{pool}$ is the average pooling operation.

The spatial layer is used to extract the relationships in the internal space and calculate which small piece should be focused on. The process is shown in Figure 7, where the feature map is convolved through a 7 × 7 2-dimensional convolutions, after which it is fed into the sigmoid layer to compute the spatial attention map.

### 3.5. Visualization of Pavement Defective Regions

The use of classification models enables one to obtain enough classification accuracy, but the use of evaluation criteria alone does not allow one to judge the results intuitively. It tries to visualize the defect region and use it to judge the advantage of RGB-thermal as input to the results. We use the gradient-weighted class activation mapping (Grad-CAM) visualization method, which can be used to visualize defects without changing the structure. Grad-CAM solves the back-propagation vanishing gradient problem by partial differentiation. Grad-CAM is summarized in Equations (3)–(5), where $a_{k}^{c}$ is the weight of layer A at channel k. This is calculated by using backpropagation with the prediction fraction $y^{c}$ of category c at layer A. The gradient information back-propagated to layer A is then used to calculate the importance of each channel k of feature layer A. The data from each channel of feature layer A are weighted and summed. Finally, the Grad-CAM is obtained by the ReLU activation function.
(3)$$a_{k}^{c}=\frac{1}{z}\sum_{i}\sum_{j}\frac{\partial y^{c}}{\partial A_{ij}^{k}}$$
(4)$$\mathrm{ReLU}=\max\left(0,x\right)$$
(5)$$L_{\mathrm{Grad}-\mathrm{CAM}}^{c}=\mathrm{ReLU}\left(\underset{k}{\sum }a_{k}^{c}A^{k}\right)$$

## 4. Experiments Sitting

### 4.1. Implementation Details of the Experiments

The training software environment is based on the pytorch platform. The operating system is Windows 10. All the experiments were conducted at a workstation equipped with an Intel 10400 6-Core CPU, 32 GB memory, and Nvidia v100 32 GB CPU card. Depending on the convergence of the accuracy curves of the training and validation sets, the early stopping method is used to prevent overfitting. After each iteration, the relevant parameters are automatically saved. An adaptive learning rate algorithm is used to automatically adjust the learning rate according to the convergence of the accuracy curve. The main objective of this study is to investigate the classification performance of a deep learning network based on a hierarchical residual network and attention mechanism for multiple pavement defects and compare it with VGG 19, ResNet50 and Inception V3. The dataset was divided into a training set, test set, and validation set at a ratio of 6:2:2.

To maximize the performance of the experimental setup and overcome the problem of the large data volume, we set the batch size value to 64. In training, the training data are iteratively fitted with suitable network parameters. As the number of epochs increases, the number of iterations to update the weights in the neural network increases, and the curve gradually moves from the initial unfitted state to the optimized fit, and the training automatically saves the optimal model. Our dataset has nine pavement defect types, which is a multi-classification problem. We use the cross-entropy loss function. For the multiclassification problem, the derivation is simple and is only related to the probability of the correct category. Since the Adam optimizer is good at handling sparse gradients and non-smooth targets, has small memory requirements, and is suitable for large datasets, we use the Adam optimizer for model training with the learning rate set to 0.001 to accelerate the network convergence.

### 4.2. Evaluation Metrics

To assess the accuracy of the results produced by the models considered, the following set of evaluation criteria are considered in our investigation. True Positive (*TP*) and True Negative (*TN*) represent the number of correctly predicted pavement defect types, respectively. False Positives (*FP*) and False Negatives (*FN*) represent the number of incorrectly predicted pavement defect types. More specifically, we use the Accuracy, Precision, Recall, and $F1_{score}$ to evaluate the model.
(6)$$Accuracy=\frac{TP+TN}{TP+FP+TN+FN}$$
(7)$$Recall=\frac{TP}{TP+FN}$$
(8)$$Precision=\frac{TN}{TN+FP}$$
(9)$$F1_{score}=2\times \frac{Precision\times Recall}{Precision+Recall}$$

## 5. Results and Analysis

### 5.1. Performance Comparison of the Proposed Model

To investigate the classification performance of the proposed RGB-thermal fusion classification model, and also evaluate the effectiveness of the different input images (RGB or RGB-Thermal images) compared to the proposed model, the comparison of the performance is listed in Table 3. To achieve these comparisons, for model comparison, it includes a comparison with existing mainstream classification networks based on RGB images as input. For the comparison of the effectiveness of the attention mechanism, this part attempts to analyze the effect of using, as well as not using, attention in the proposed network on the classification rate of pavement defects. The effectiveness of the different input images was also compared by using different input images in the proposed model.

The experimental results show that the testing accuracy of the proposed RGB-thermal fusion classification model with the attention mechanism using RGB-thermal images is 98.88% for the classification of multiple types of pavement distresses, which achieved the best accuracy of all comparison models. When the attention module is not embedded in the proposed model, likely due to the specificity of different sensor images, the classification accuracy of the test set is reduced to 94.74%. This shows that the attention mechanism can effectively extract effective features by adjusting the weights of channels as well as spatial. Weight-adjustment-based fusion for combinations between multiple sensors contributes to the final classification accuracy.

In addition, we trained and compared other image classification networks. Among them, the test set accuracies of VGG 19, ResNet50, and Inception V3 were 95.33%, 96.63%, and 96.94%, respectively. It is worth noting that since these networks are proposed based on RGB images, only RGB images are used here as input. The final testing accuracy of all models using RGB as inputs is approximately 96%. This also shows that the existing models using RGB as input are able to achieve relatively good prediction results. The final accuracy obtained is not very different. Meanwhile, the proposed network with the attention mechanism using RGB images does not effectively improve the accuracy, as it improved the testing accuracy from 95.94% to 96.26%.

Comparing the training loss curves (shown in Figure 8) of all models, it is found that the proposed RGB-thermal fusion classification model with the attention mechanism (proposed_rgbt-at in Figure 8) has a poor network fitting ability in the first five epoch periods. With the continuous training of the network, the fitting ability of the model gradually improved. The network converges and is stable. Compared to the proposed model without the attention mechanism (proposed_rgbt shown in Figure 8), it converges at the same rate, but the final validation accuracy is lower than the proposed_rgbt-at. In contrast, proposed_rgb and proposed_rgb-at, which only use RGB as the input, converge faster during training but achieve lower validation accuracy than proposed_rgbt-at during validation. This indicates that when using rgb-thermal images as input, more useful features for classification can be input, and the attention mechanism can cull these input features from both RGB and thermal images to achieve better prediction accuracy.

Since there is a lack of existing classification models using rgb-thermal as the input, we compare models that also use RGB as input. As we can see from the training convergence speed, both proposed_rgbt and proposed_rgb-at are faster than inception V3, ResNet V3, and Vgg19. Even the proposed_rgbt-at, which uses rgbt as the input, converges faster than inception V3, ResNet V3, and Vgg19 in the training stage. It shows that the proposed model is able to guarantee better prediction accuracy with faster training convergence.

The running efficiency of the model is also a criterion for the goodness of the model. Table 4 shows the model size and training time. We compare all models in terms of model size and training time. The total training time is the time it takes for the model to start training and finish, while the time per calendar time is the total training time divided by the total calendar time. The experimental results show that the proposed model (RGB without AM) performs best in terms of the model size and training time, with a model size of 53.66 MB and a training time of 41.76 min. When RGB-thermal is used as the input, the model size, training time, and inference time increase, with a model size of 55.24 MB. It should be mentioned here that rgb-thermal has a longer read time for a 4-channel tif file format. Referring to Table 4, it can be seen that our model outperforms InceptionV3, ResNet50, and vgg19 in terms of model size, training time, and inference time. The study shows that our model has a smaller size, shorter training time, and higher accuracy.

### 5.2. Classification Results of the Proposed Model

We used the classification results of the proposed model to compare the results of the comparison model to verify the effectiveness of the model in classifying single pavement defects. Figure 9 shows the use of the proposed rgbt-at model to identify various pavement damage images. The experimental results show that the overall testing accuracy of the rgbt-at model for pavement defects is 98.88%. The model performs best in classifying manholes, with an accuracy of 100%. The model was less accurate in classifying alligator cracks, with an accuracy of 98.17%. It would incorrectly predict these as a longitudinal crack, oil marks, or transverse crack. The chances of confusing oil marks, shadows, and potholes were high. The reason for this may be that oil marks, shadows, and pothole features do not differ much and have relatively similar texture, color, and temperature characteristics, so the accuracy is low.

Figure 10 compares the specific accuracies of different networks for different pavement defects and identifiable markers. The networks using RGB images as input have good recognition performance, except for vgg19, which performs slightly below average. However, the network using rgbt as input without the attention mechanism instead leads to lower recognition accuracy. The prediction accuracy of each of the compared networks showed the same trend as the average accuracy). In conclusion, the rgbt-at network with feature and channel selection based on the attention mechanism was able to achieve better prediction accuracy when using rgbt as a redundant information source as input.

### 5.3. Visualization and Interpretation for the Multi-Sensor Fusion Classification Model

Figure 11, Figure 12 and Figure 13 show the Grad-CAM outputs of the networks using RGB images, thermal images, and RGB-thermal images as input, respectively. As shown in the figures, the input sample images, the output map of Grad-CAM, and the superimposed images of Grad-CAM with the input samples are shown, respectively. In the Grad-CAM heat map, the brighter colors indicate the regions that contribute more to the classification results in the heat map. Correspondingly, in the Grad-CAM output image, the yellow area indicates a higher score for that class. As can be seen in Figure 11, Grad-CAM is able to capture the approximate area where the classification is needed. The best results are obtained for potholes and oil marks. On the contrary, the cracks (including alligator, longitudinal cracks, and transverse cracks) presented in the RGB images are relatively small and cannot be captured well in the grad-cam maps. They express color as well as texture information, while thermal mainly expresses temperature information. As can be seen in Figure 12, there is a clear difference in the crack region (the temperature difference is expressed as a grayscale difference in the pixel values), and the features in the crack region have a clearer Grad-CAM image. However, using the thermal image as input compared to the Grad-CAM image results in poorer performance in capturing oil marks and the pothole. The Grad-CMA image of the fused image is shown in Figure 13, and the boundary of the area to be classified becomes more detailed and clearer. In a way, it explains why fused image classification can achieve better classification accuracy. Overall, from the interpretation of the results, we can find that the final trained network will focus more on the relevant parts of the image. The more relevant parts are extracted, and the more explicit they are, the better the prediction performance will be.

## 6. Discussion

In this paper, we propose a new model for pavement defect classification using RGB-thermal images. Furthermore, we explore the effect of the attention mechanism and different input data (rgb and rgn-thermal) on the classification performance of the model. The experimental results show that the model using rgb-thermal images as input improves the classification performance of plant diseases, especially when the improved attention mechanism is embedded in the model.

In this study, although the dataset was collected with a large number of pavement damage images, the images were taken under controlled conditions, and only one pavement defect or object to be detected was present in each image. This is the ideal image acquisition condition, which has more limitations in practice, such as the infrared camera’s imaging being affected by the outside temperature. This is reflected in the fact that the applicability of the proposed method is unknown for complex weather conditions. We are unable to collect data in the short term for a wide range of weather and temperature conditions. Even due to these limitations, our proposed method presents a new idea for the classification monitoring of pavement defects, which can be further explored in future developments for applicability in complex weather conditions. Although our proposed model has drawbacks in the acquisition of experimental data, our model has many advantages; specifically, our model has higher recognition accuracy compared to existing techniques, and our model is more lightweight with fewer parameters and shorter training time.

## 7. Conclusions

In this paper, a convolutional neural network for the classification of thermal images using RGB images is proposed. The classification network based on the improved hierarchical residual module with a simplified network structure can reduce the training parameters, shorten the training time, and improve the training stability. The hierarchical residual module, which can extract multi-scale feature information by concatenating multiple convolutional layers with different combinations, as well as the residual block, can also solve the degradation problem of deep networks. To verify the performance of the model and the effect of different input images on the prediction results, our model has higher operational efficiency and recognition accuracy with 96.99% accuracy, when RGB images are used as input, compared to InceptionV3, ResNet50, and Vgg19. Using the proposed model with RGB-thermal and RGB as input, RGB-thermal can achieve better model prediction performance with 98.88% accuracy. It is also verified that after using the attention mechanism, it is possible to adjust the trade-off of redundant input information when multiple sensors are used as inputs. Accuracies of 95.24% and 98.88% were obtained without and with the attention mechanism, respectively.

## Figures and Tables

**Figure 1 sensors-22-05781-f001:**
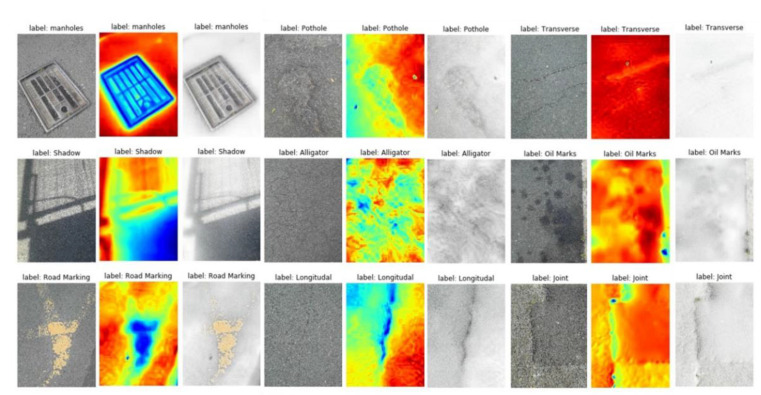
RGB, thermal, and fusion image samples of the pavement defects and detectable markers.

**Figure 2 sensors-22-05781-f002:**
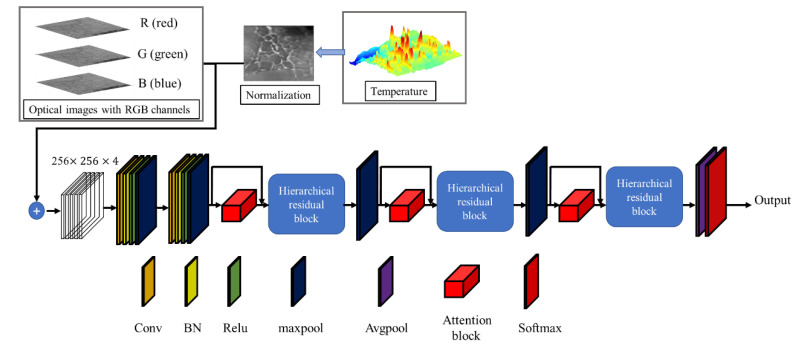
Framework of proposed model based on hierarchical residual network and attention mechanism.

**Figure 3 sensors-22-05781-f003:**
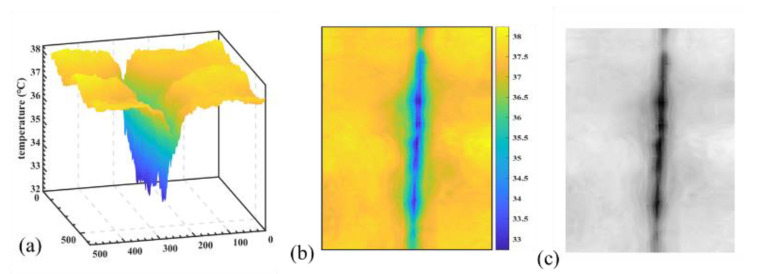
(**a**) The 3D schematic of the thermal image, (**b**) thermal image, (**c**) and transformed temperature grayscale image.

**Figure 4 sensors-22-05781-f004:**
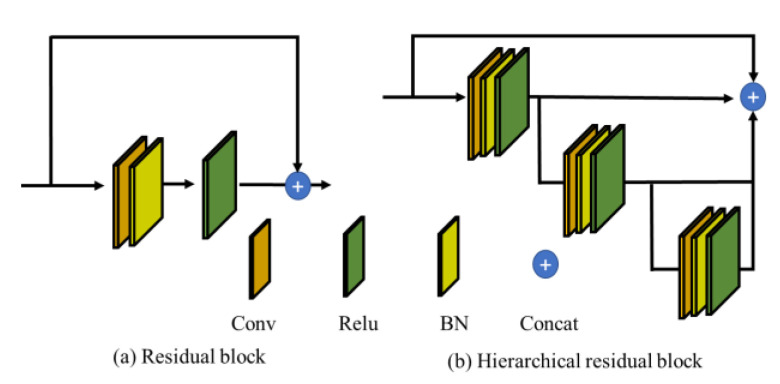
(**a**) Illustration of the residual block; (**b**) illustration of the hierarchical residual block.

**Figure 5 sensors-22-05781-f005:**
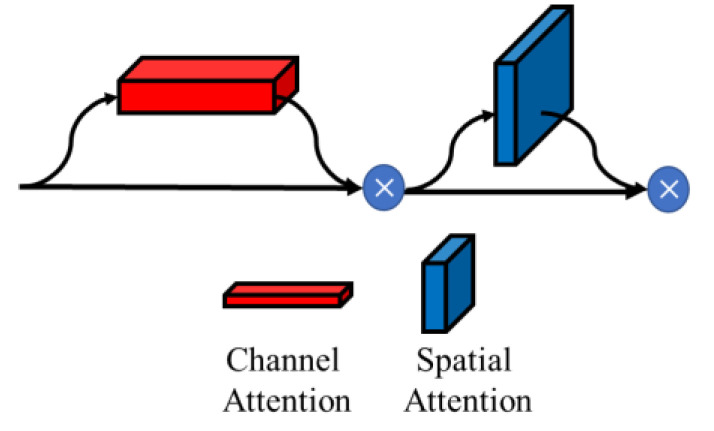
CBAM (Convolutional Block Attention Module).

**Figure 6 sensors-22-05781-f006:**
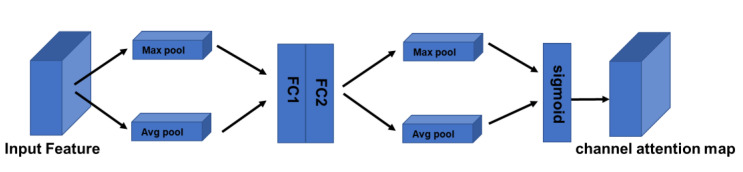
Illustration of channel attention operation.

**Figure 7 sensors-22-05781-f007:**
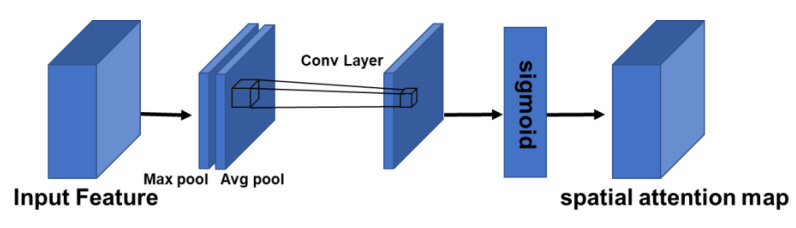
Illustration of the spatial attention operation.

**Figure 8 sensors-22-05781-f008:**
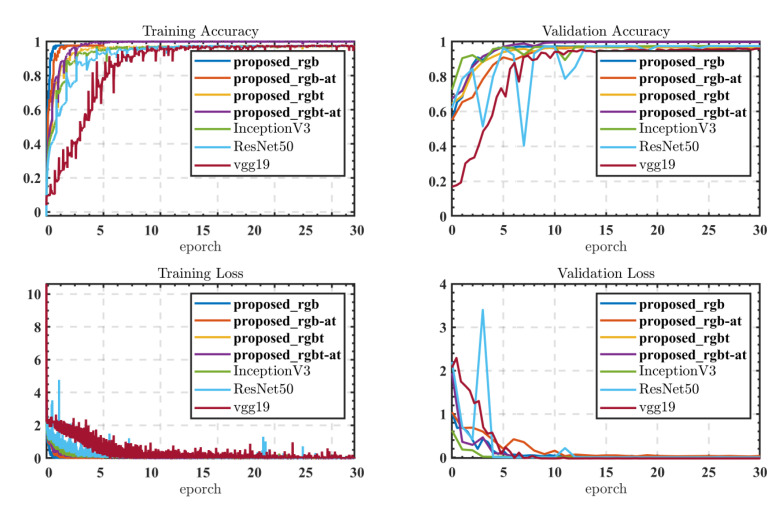
Training accuracy and loss curve of the models.

**Figure 9 sensors-22-05781-f009:**
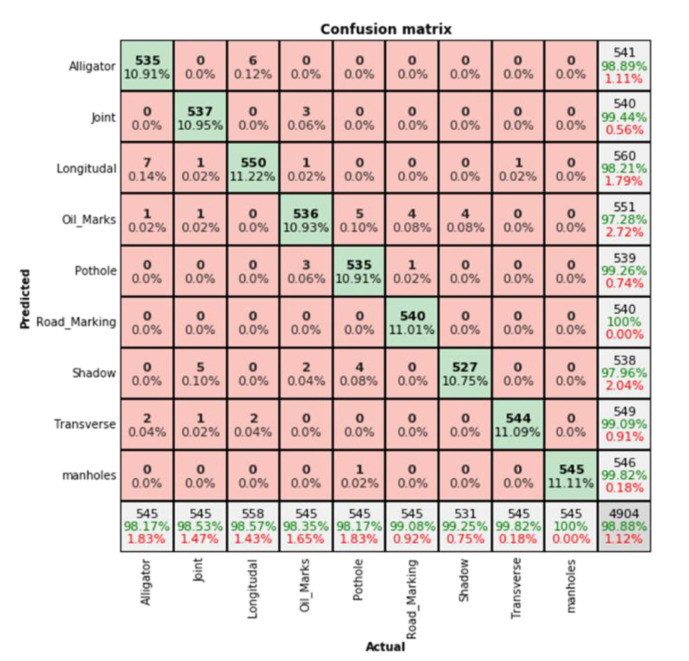
Confusion matrix of pavement defects of the proposed rgbt-at model.

**Figure 10 sensors-22-05781-f010:**
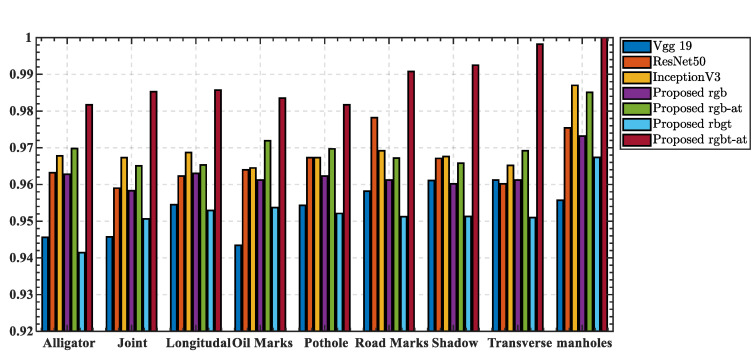
Detailed comparison results of prediction rate of different models.

**Figure 11 sensors-22-05781-f011:**
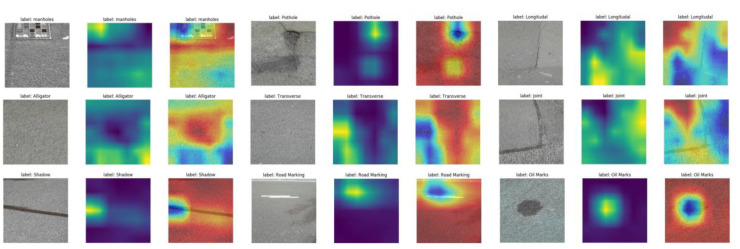
Visualization of the RGB images as inputs using Grad-CAM.

**Figure 12 sensors-22-05781-f012:**
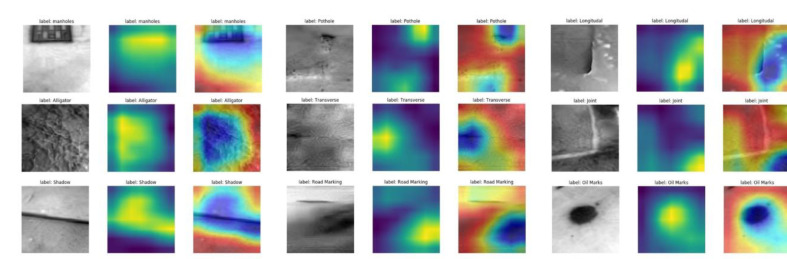
Visualization of the thermal images as inputs using Grad-CAM.

**Figure 13 sensors-22-05781-f013:**

Visualization of the RGB-thermal images as inputs using Grad-CAM.

**Table 1 sensors-22-05781-t001:** Description of dataset.

Distress Types	Transverse Cracks	Longitudinal Cracks	Alligator Cracks	Joints or Patches	Potholes	Manholes	Shadows	Road Markings	Oil Markings
amount	2745	2800	2705	2700	2695	2730	2690	2700	2755

**Table 2 sensors-22-05781-t002:** Proposed model parameters.

Layer	Kernel	Stride	Output Shape
Input			(None,4,256,256)
Conv2d 1	3 × 3	2	(None,64,128,128)
BN 1 and ReLU			(None,64,128,128)
Max pool 1	2 × 2		(None,64,64,64)
Conv2d 2	3 × 3	1	(None,128,64,64)
BN 2 and ReLU			(None,128,64,64)
Max pool 2	2 × 2		(None,128,32,32)
attention 1			(None,128,32,32)
HR-Block 1			(None,128,32,32)
Max pool 3	2 × 2		(None,128,16,16)
attention 2			(None,128,16,16)
HR-Block 2			(None,128,16,16)
Max pool 4	2 × 2		(None,128,8,8)
attention 3			(None,128,8,8)
HR-Block 3			(None,128,8,8)
average pool	2 × 2		(None,128)
SoftMax			(None,9)

**Table 3 sensors-22-05781-t003:** Data results of different models’ training and testing.

Model	Training Accuracy %	Training Loss	Testing Accuracy %	Testing Loss	Precision %	Recall %	F1-Score %
Vgg 19	98.23	0.0488	95.33	0.023	94.37	95.12	94.74
ResNet50	98.46	0.0496	96.63	0.045	96.47	97.01	96.74
InceptionV3	98.17	0.0469	96.94	0.32	96.01	96.21	96.11
Proposed model (RGB without AM)	98.26	0.0453	96.26	0.0512	96.32	96.27	96.29
Proposed model (RGB with AM)	98.86	0.0473	96.99	0.0573	97.01	96.46	96.73
Proposed model (RGBT without AM)	96.32	0.083	95.24	0.1123	95.54	94.20	94.87
Proposed model (RGBT with AM)	99.12	0.0451	98.88	0.0452	99.12	98.21	98.66

**Table 4 sensors-22-05781-t004:** Model size and training time.

Model	Size (MB)	Time per Epoch (min)	Total Training Time (min)
Vgg 19	845.05	5.78	578
ResNet50	432.21	3.84	384
InceptionV3	300.54	3.11	311
Proposed model (RGB without AM)	53.66	1.044	41.76
Proposed model (RGB with AM)	55.24	1.032	41.28
Proposed model (RGB-Thermal without AM)	53.91	1.89	75.6
Proposed model (RGB-Thermal with AM)	55.24	1.9	76

## Data Availability

Some or all data, models, or code generated or used during the study are available from the corresponding author by request.

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
