# Peer review of "Automatic Pavement Defect Detection and Classification Using RGB-Thermal Images Based on Hierarchical Residual Attention Network"

_sensors, 2022, doi:10.3390/s22155781_

Round 1
Reviewer 1 Report
In this paper, a convolutional neural network based on an improved residual structure is proposed to implement a classification model for recognizing complex pavement conditions. The structure is reasonable and the results are reliable. However, there are some little suggestions and comments for authors:
1) The first part of introduction should seperate the background from previous studies.
2) The abbreviation in the abstract should be supplemented with the full name,e.g. SOTA.
3) Although this paper exemplifies previous studies, it does not state the differences between this paper and previous studies,which should be added before line 74.
Author Response
Thanks for the comments of the reviewer and the detailed responses is attached.

Reviewer 2 Report
Thank you to the authors for submitting their work for consideration for publication. In this paper, the authors developed a classification scheme for asphalt pavement distresses using thermal imaging systems. Overall, the methodology seems sound and the paper is interesting. I just have a few comments for the authors to consider:
1. The English should be checked and revised.
2. I'm not sure about the title's use of "defection." Do the authors mean "defect detection"?
3. Line 78 - The authors say nine pavements were considered, but I think they mean nine distresses. If this is the case, the authors need to describe the number of pavement sections and how many of each distress were in the data set.
4. Many papers investigated this problem before- please check the literature and include a wider range of studies on defect detection and how they solved the problem.
5. Lines 250 - 251 - How did the authors check to ensure overfitting was prevented?
Author Response
Thanks for the comments of the reviewer and the detailed responses are attached.

Reviewer 3 Report
This paper describes a deep neural network to detect pavement defects.
The paper is not easy to read. There are sentences that should be rephrased and typos. Thus, a carefully revision of the English style should be performed. Moreover, at the end of the Introduction the authors should describe the paper organization.
The dataset is not described in details. How many images have been acquired? How many images belong to each class? How have the images been labelled?
Finally, the proposed method should be compared with other solution from the literature, not only with other networks developed by the authors.
Author Response

(The authors gave the same response as above.)

Round 2
Reviewer 3 Report
Authors addressed all my comments.